

# Retrospective study of prognostic factors in pediatric invasive pneumococcal disease

Nan-Chang Chiu[1,2,3,*], Hsin Chi[1,2,3,*], Chun-Chih Peng[1,2,3], Hung-Yang Chang[1,4], Daniel Tsung-Ning Huang[1,2], Lung Chang[1,2], Wei-Te Lei[5] and Chien-Yu Lin[5]

[1] Department of Pediatrics, MacKay Children's Hospital, Taipei, Taiwan
[2] MacKay Junior College of Medicine, Nursing and Management, Taipei, Taiwan
[3] MacKay Medical College, New Taipei, Taiwan
[4] Department of Medical Technology, Jen-Teh Junior College of Medicine, Nursing and Management, Miaoli, Taiwan
[5] Hsinchu MacKay Memorial Hospital, Hsinchu, Taiwan
[*] These authors contributed equally to this work.

Corresponding author
Chien-Yu Lin,
mmhped.lin@gmail.com

## ABSTRACT

*Streptococcus pneumoniae* remains the leading causative pathogen in pediatric pneumonia and bacteremia throughout the world. The invasive pneumococcal disease (IPD) is known as isolation of *S. pneumoniae* from a normally sterile site (e.g., blood, cerebrospinal fluid, synovial fluid, pericardial fluid, pleural fluid, or peritoneal fluid). The aim of this study is to survey the clinical manifestations and laboratory results of IPD and identify the prognostic factors of mortality. From January 2001 to December 2006, a retrospective review of chart was performed in a teaching hospital in Taipei. The hospitalized pediatric patients with the diagnosis of pneumonia, arthritis, infectious endocarditis, meningitis or sepsis were recruited. Among them, 50 patients were pneumococcal infections proved by positive culture results or antigen tests. Clinical manifestations, laboratory data and hospitalization courses were analyzed. The median age was 3.5-year-old and there were 30 male patients (60%). Eight patients (16%) had underlying disease such as leukemia or congenital heart disease. Hemolytic uremic syndrome (HUS) was observed in ten patients and extracorporeal membrane oxygenation (ECMO) was performed in three patients. Leukocytosis, elevated C-reactive protein and AST level were noted in most of the patients. The overall mortality rate was 10%. We found that leukopenia, thrombocytopenia and high CRP level were significant predictors for mortality. In conclusion, *S. pneumoniae* remains an important health threat worldwide and IPD is life-threatening with high mortality rate. We found leukopenia, thrombocytopenia, and high CRP levels to be associated with mortality in pediatric IPD, and these factors are worthy of special attention at admission. Although we failed to identify a statistically significant prognostic factor in multivariate analysis due to relatively small sample size, we suggest an aggressive antibiotic treatment in patients with these factors at admission. Further large-scale studies are warranted.

## INTRODUCTION

*Streptococcus pneumoniae*, a Gram-positive diplococcus, remains the leading causative pathogen in pediatric pneumonia and bacteremia worldwide (*Jroundi et al., 2014*; *Tan, 2012*). Colonization with *S. pneumoniae* is common and contributes to invasive disease (*Hamaluba et al., 2015*). In 2000, *S. pneumoniae* reportedly caused 14.5 million episodes of serious disease and 11% of all deaths in children younger than 5 years (*O'Brien et al., 2009*). Invasive pneumococcal disease (IPD) is defined as isolation of *S. pneumoniae* from a normally sterile site (e.g., blood, cerebrospinal fluid (CSF), synovial fluid, pericardial fluid, pleural fluid, or peritoneal fluid) (*Levine et al., 1999*). The mortality rate associated with IPD is high, ranging from approximately 5.3–27.5%, and identification of high-risk groups is important (*Berjohn et al., 2008*; *Chen et al., 2009*; *Gomez-Barreto et al., 2010*; *Marrie et al., 2011*; *Ruckinger et al., 2009*; *Rueda et al., 2010*; *Shariatzadeh et al., 2005*; *Ulloa-Gutierrez et al., 2003*). Some factors associated with outcome include breastfeeding, passive smoking, antimicrobial resistance, pneumococcal serotype, and prompt use of antibiotics (*Berjohn et al., 2008*; *Haddad et al., 2008*; *Ruckinger et al., 2009*; *Tan, 2012*). The emerging resistance of *S. pneumoniae* brings challenges to successful treatment (*Ibrahim et al., 2000*; *Minami et al., 2015*). The aim of this study was to survey the clinical manifestations and laboratory results of IPD in children and identify the prognostic factors of mortality.

## METHODS

### Study design and study population

This study was approved by the Ethics Committees of MacKay Memorial Hospital, Taiwan (IRB No.: 16MMHIS035e). MacKay Memorial Hospital is a tertiary teaching hospital in Taipei, Taiwan. From January 2001 to December 2006, a retrospective chart review of hospitalized pediatric patients were performed. Patients younger than 18 years old hospitalized for infectious etiologies, eg. pneumonia, meningitis, or sepsis, were enrolled in the study. Patients admitted for noninfectious causes were excluded, such as preterm delivery, diabetes mellitus, epilepsy, cardiovascular diseases, or scheduled chemotherapy. Since chronic underlying illness is a risk factor of IPD, patients with underlying diseases were not excluded in our analysis. The clinical history and laboratory tests were investigated to identify those with IPD. IPD is also a communicable disease in Taiwan and physicians have to report to Centers for Disease Control, Taiwan about patients with IPD. We also search the database of communicable diseases to detect patients with IPD. The electronic charts were also reviewed using the ICD-9-CM coding system to reduce missing patients with IPD. Only laboratory-confirmed cases were enrolled for investigation and analysis.

In total, 13,141 patients were admitted for infectious etiologies during the study period and enrolled in our screen. Among them, 50 (0.4%) patients with pneumococcal infections proven by positive culture results or antigen tests were detected. Five patients (10%) died and the clinical manifestations, laboratory data, and hospitalization courses were compared between the survival and mortality groups.

## Laboratory tests

Upon admission, blood tests including complete blood count, C-reactive protein (CRP) and blood cultures were routinely obtained. Spinal tap, pleurocentesis and video-assisted thoracoscopic surgery were selectively performed according to the patients' clinical conditions and the judgement of physicians. Data from blood, CSF, and pleural fluid cultures performed using aseptic technique according to manufacturer's instructions were reviewed. Urine pneumococcal antigen, a reliable tool with a moderate sensitivity and good specificity to detect *S. pneumoniae*, was also investigated (Binax NOW *S. pneumoniae* urinary antigen test; Binax, Portland, Maine) (*Roson et al., 2004*). Complete blood count, CRP, Thomsen-Friedenreich antigen, and biochemistry results were also reviewed. Virus studies were also performed and analyzed. Antibiotic susceptibility testing interpretation was based on Clinical and Laboratory Standards Institute 2010 criteria.

## Statistical analysis

The statistical analysis was performed using SPSS version 15.0 (SPSS, Chicago, IL). Categorical variables were compared using the chi-square test and Fisher's exact test. Continuous variables were compared using the t-test. Variables with a $p$-value $< 0.05$ were included into multivariate analysis. Log regression was performed for multivariate analysis. A $p$-value $< 0.05$ was considered statistically significant.

## RESULTS

Of the 50 patients (median age, 3.5 years) included in the study, 30 (60%) were males (Table 1). Eight patients (16%) had underlying diseases, such as leukemia or congenital heart disease. IPD was more common in cold seasons (36 patients, 72% in autumn and winter) and pneumonia was diagnosed in 46 patients (92%). Twenty-eight patients (56%) had sepsis and four (8%) had meningitis. Three patients had concurrent involvement of pneumococcal meningitis, sepsis, and pneumonia. Concurrent virus infection was noted in two patients (echovirus 6 and respiratory syncytial virus, respectively) and both of them survived. Hemolytic uremic syndrome (HUS) was observed in ten patients and extracorporeal membrane oxygenation (ECMO) was performed in three patients. Leukocytosis and elevated levels of CRP and aspartate aminotransferase were noted in most of the patients. Median hospital stay was 15 days and the overall mortality rate was 10%.

Comparing the clinical manifestations, laboratory data, and hospital course between the survival and mortality groups, we found that leukopenia (white blood cell $< 4,000/\mu L$), thrombocytopenia (platelet count $< 100,000/\mu L$), and high CRP levels (CRP $> 20$ mg/dL) were significant predictors for mortality (Table 2). Antibiotic resistance was not associated with mortality. The multivariate analysis showed no significant single prognostic factor.

## DISCUSSION

*S. pneumoniae* is the most important bacterial pathogen in both invasive and mucosal disease worldwide (*Jroundi et al., 2014*; *O'Brien et al., 2009*; *Tan, 2012*; *Wei et al., 2015*). In 2000, it was estimated to cause 14.5 million episodes of serious disease and 11% of all
**Table 1 Clinical manifestations of patients with IPD.**

|  | Overall ($n = 50$) | Survival ($n = 45$) | Mortality ($n = 5$) | $p$-value |
|---|---|---|---|---|
| Age, y | $3.5 \pm 0.82$ | $3.6 \pm 0.92$ | $2.7 \pm 1.5$ | 0.083 |
| Male, $n$ (%) | 30 (60) | 26 (57.8) | 4 (80) | 0.63 |
| Cold seasons, $n$ (%) | 36 (72) | 34 (75.6) | 2 (40) | 0.248 |
| Underlying disease, $n$ (%) | 8 (16) | 7 (15.6) | 1 (20) | 0.70 |
| Involved organ |  |  |  |  |
| Lung, $n$ (%) | 46 (92) | 42 (93.3) | 4 (80) | 0.862 |
| Blood, $n$ (%) | 28 (56) | 27 (60) | 1 (20) | 0.217 |
| CNS, $n$ (%) | 4 (8) | 2 (4.4) | 2 (40) | 0.056 |
| Hospital days, d | 14.56 | $15 \pm 2.76$ | 10.2 |  |
| ECMO use, $n$ (%) | 3 (6) | 1 (2.2) | 2 (40) | **0.017*** |
| HUS, $n$ (%) | 10 (20) | 9 (20) | 1 (20) | 0.72 |
| Laboratory tests |  |  |  |  |
| Hb, g/dl | $11.37 \pm 0.48$ | $11.92 \pm 2.03$ | $11.3 \pm 0.5$ | 0.441 |
| WBC, /$\mu$L | $16,004 \pm 2,861$ | $16,824 \pm 2,966$ | $8,624 \pm 5,227$ | 0.084 |
| Platelet, /$\mu$L | $311.8 \pm 54.9$ | $322.7 \pm 56.7$ | $215.6 \pm 283.5$ | 0.239 |
| CRP, mg/dl | $17.95 \pm 4.28$ | $16.63 \pm 4.61$ | $29.06 \pm 6.99$ | 0.071 |
| AST, IU/L | $70.25 \pm 29.23$ | $62.26 \pm 31.5$ | $100.6 \pm 102.94$ | 0.28 |
| Creatinine, mg/dl | $0.86 \pm 0.28$ | $0.9 \pm 0.36$ | $0.74 \pm 0.45$ | 0.667 |
| Na, meq/L | $135.3 \pm 1.6$ | $135.5 \pm 1.7$ | $134.4 \pm 5.9$ | 0.599 |
| Resistant to Penicillin | 16 (37.2) | 15 (37.5) | 1 (33) | 0.63 |

**Notes.**
For categorical variables, the results are expressed as number (percentage); for continuous variables, the results are expressed as mean $\pm$ standard deviation. A $p$-value less than 0.05 is considered statistically significant and indicated by an asterisk (*).

**Table 2 Prognostic factors of patients with IPD.**

|  | Overall ($n = 50$) | Survival ($n = 45$) | Mortality ($n = 5$) | $p$-value |
|---|---|---|---|---|
| Age $\leq 2$ y, $n$ (%) | 17 (34) | 15 (33.3) | 2 (40) | 0.842 |
| Hb $\leq 10$ g/dL, $n$ (%) | 9 (18) | 8 (17.8) | 1 (20) | 0.624 |
| WBC $\leq 4,000$ /$\mu$L, $n$ (%) | 7 (14) | 4 (8.9) | 3 (60) | **0.015*** |
| WBC $\geq 20,000$ /$\mu$L, $n$ (%) | 15 (30) | 14 (31.1) | 1 (20) | 1.00 |
| Plt $\leq 100,000$ /$\mu$L, $n$ (%) | 7 (14) | 4 (8.9) | 3 (60) | **0.014*** |
| CRP $\geq 20$ mg/dL, $n$ (%) | 22 (44) | 17 (37.8) | 5 (100) | **0.029*** |
| Meningitis, $n$ (%) | 4 (8) | 2 (4.4) | 2 (40) | **0.056** |
| ECMO use, $n$ (%) | 3 (6) | 1 (2.2) | 2 (40) | **0.017*** |

**Notes.**
For categorical variables, the results are expressed as number (percentage); for continuous variables, the results are expressed as mean $\pm$ standard deviation. A $p$-value less than 0.05 is considered statistically significant and indicated by an asterisk (*).

deaths in children younger than five years (*O'Brien et al., 2009*). The carrier rate is high and is influenced by a number of factors, including age, race, exposure to cigarette smoke, and day care attendance (*Ebruke et al., 2015*; *Tan, 2012*). In a cross-sectional observational study conducted in the United Kingdom, the carrier rate in children was 47% (*Hamaluba et al., 2015*). High prevalence of nasopharyngeal carriage were also observed in many areas, such as Taiwan (21%) and Egypt (29.2%) (*Chiou et al., 1998*; *El-Nawawy et al., 2015*).

**Table 3  Reported mortality in patients with IPD.**

| Authors | Ulloa-Gutierrez | Ruckinger | Gomez-Barreto | Shariatzadeh | Marrie | Rueda | Berjohn | Chen | Current study |
|---|---|---|---|---|---|---|---|---|---|
| Age | Children | Children | Children | Adults | Adults | Adults | Adults | Children | Children |
| Study period | 1995–2001 | 1997–2003 | 1997–2004 | 2000–2002 | 2000–2004 | 2000–2008 | 2001–2004 | 2001–2006 | 2001–2006 |
| Country | Costa Rica | Germany | Mexico | Canada | Canada | United States | United States | Taiwan | Taiwan |
| IPD | All IPD | All IPD | All IPD | Pneumonia, bacteremia | All IPD | All IPD | Pneumonia, bacteremia | All IPD | All IPD |
| Mortality (%) | 14.4 | 5.3 | 27.5 | 9.3 | 14.1 | 16.2 | 10 | 8.1 | 10 |

Followed by the high prevalence of nasopharyngeal carriage, *S. pneumoniae* is the most common pathogen in children with severe infection (*Jroundi et al., 2014*). In hospitalized children with acute respiratory infections, *S. pneumoniae* was the most frequently detected bacteria (14.4%) (*Wei et al., 2015*). In patients with IPD, the mortality rate is high in both children and adults and the findings of some important studies are summarized in Table 3. Most of IPD developed in young children and our study is consistent with previous report (*Centers for Disease Control and Prevention, 2010*). Only two patients were older than six years old and both of them survived. Although the mortality rate varies in different countries, time, and study, IPD remains an important health issue. Prompt diagnosis and timely treatment are important. The burden of disease is heavy and *S. pneumoniae* remains the most important health threat worldwide.

Pneumococcal pneumonia with empyema is the most common form of IPD. Infectious endocarditis is a rare form of IPD, with a mortality rate 20.7% (*De Egea et al., 2015*). Most cases of infectious endocarditis occur in adults with underlying disease; there were no children with pneumococcal infectious endocarditis in our study group. The presence of pneumococcal carriage and IPD is multifactorial. The presence of an underlying chronic illness is a risk factor, whereas breastfeeding has protective effect against IPD (*Chen et al., 2009*; *Haddad et al., 2008*; *Lee et al., 2003*; *Levine et al., 1999*). Race and group child care attendance also play a role in IPD (*Pilishvili et al., 2010*). In patients with IPD, prompt and timely use of antibiotics is associated with favorable outcome; apparently, antimicrobial resistance does not affect outcome (*Berjohn et al., 2008*; *Kumashi et al., 2005*). The findings in our study are consistent with those in previous reports. Upon admission, the presence of cytopenia and meningitis may indicate unfavorable outcome (*Chen et al., 2009*). In addition, we found thrombocytopenia and high CRP levels to be associated with mortality. ECMO use and the presence of meningitis were indicative of more severe disease and were associated with poor outcome. The presence of HUS was not suggestive of poor prognosis in our study. Increased resistance of *S. pneumoniae* has been reported and prompt and adequate antimicrobial treatment is essential for successful treatment (*Ibrahim et al., 2000*; *Minami et al., 2015*). For patients with suspected IPD, bacterial cultures are the gold standard and provide tailored guidance to antibiotics use. It is important to obtain specimens before initiating antimicrobial agents. However, cultures are time-consuming and adequate empirical antibiotics regimen remains a challenge for primary physicians.

Based on the finding of our study, we suggest an aggressive antibiotic treatment and early surgical intervention if needed in patients with leukopenia, thrombocytopenia and elevated CRP level at admission.

The host immune responses to infections are regulated by a mixture of pro-inflammatory and anti-inflammatory mediators. In addition to traditionally laboratory tests, genes encoding these mediators are associated with disease progression and outcomes of patients with sepsis and are regarded as a prognostic factor of severe infection. As medicine advances, mounting evidences show that patient-specific genetic polymorphism contributes to disease prognosis and potential for treatment (*Holmes, Russell & Walley, 2003*). Various single nucleotide polymorphisms of genes that encode cytokines, cell surface receptors, lipopolysaccharide ligands, mannose-binding lectin, heat shock protein 70, angiotensin I-converting enzyme, plasminogen activator inhibitor, and caspase-12 are associated with increased susceptibility to infection and poor outcomes (*Frantz, Ertl & Bauersachs, 2007*). For example, programmed cell death 1 protein (PD-1) is a negative costimulatory molecule and involved in the host responses to sepsis. Patients with G homozygotes of PD-1 rs11568821 had higher risk of mortality, sepsis score and a higher demand of vasopressor therapy than A allele carriers (*Mansur et al., 2014*). Similarly, CD14 plays a crucial role in initiating and perpetuating the pro-inflammatory processes during sepsis. Compared with the C-allele carrier of CD14 rs2569190, TT-homozygous patients had a favorable outcome in sepsis (*Mansur et al., 2015*). The importance of genetic polymorphism and infectious disease warrants our attention. Association between genetic polymorphisms and IPD is also reported (*Brouwer et al., 2009*; *Lundbo et al., 2016*; *Rautanen et al., 2016*). For instance, in Kenya children, polymorphism in a lincRNA was associated with a doubled risk of pneumococcal bacteremia (*Rautanen et al., 2016*). Genetic counselling was performed in our study patients but genetic susceptibilities were not tested. Further studies are warranted to elucidate the relationship between genetic polymorphism and IPD and genetic polymorphism may serve as a valuable prognostic factor in patients with IPD.

Our study has some limitations. First, the sample size was too small that the multivariate analysis showed no significant single prognostic factor. IPD is a severe and relatively rare disease and the estimated incidence of IPD was approximate 12.9/100,000 (*Centers for Disease Control and Prevention, 2010*). Our study demonstrated three prognostic factors of unfavorable outcome of IPD, but no statistically significant prognostic factor was identified in multivariate analysis due to relatively small sample size. Further large-scale, prospective studies are warranted. Second, the pneumococcal serotype was not investigated. Association between pneumococcal serotypes and patient outcomes has been reported (*Ciruela et al., 2013*; *Kapatai et al., 2016*; *Ruckinger et al., 2009*; *Weinberger et al., 2010*). However, serotype identification is not always available in most hospitals and, therefore, its clinical use is limited. Furthermore, our study was conducted before the introduction of pneumococcal conjugate vaccination in Taiwan and the protective effects of pneumococcal vaccination were not investigated. The introduction of pneumococcal conjugate vaccination has dramatically decreased the incidence of IPD (*Centers for Disease Control and Prevention (CDC), 2010*; *Abdelnour et al., 2015*; *Pilishvili et al., 2010*; *Von Gottberg et al., 2014*). In the era of post-heptavalent pneumococcal conjugate vaccine, serotype replacement and

breakthrough infection are of global concern (*Abdelnour et al., 2015*; *Park et al., 2010*; *Tan, 2012*). The vaccine coverage rate differs among different countries; for example, the coverage rate is approximately 90% in the United States and 10% in Shanghai, China (*Boulton et al., 2016*; *Hill et al., 2015*). Therefore, the protective effect of vaccination is influenced by the coverage rate and is difficult to evaluate. Finally, this study is a retrospective study and conducted in one teaching hospital in northern Taiwan. Some tests were not always available and there were some missing data. For example, spinal tap and pleurocentesis were not performed in all patients. The prevalence, managements, outcomes and findings may be different in different hospitals, areas, countries and eras.

In conclusion, *S. pneumoniae* remains an important health threat worldwide, and IPD is a life-threatening disease with a high mortality rate. Pneumococcal vaccination should be encouraged and physicians should maintain high alertness toward patients with IPD. We found leukopenia, thrombocytopenia, and high CRP levels to be associated with mortality in pediatric IPD, and these factors are worthy of special attention at admission. Although we failed to identify a statistically significant prognostic factor in multivariate analysis due to relatively small sample size, we suggest an aggressive antibiotic treatment in patients with these factors at admission. Further large-scale studies are warranted.

### Abbreviations

| | |
|---|---|
| **AST** | Aspartate aminotransferase |
| **CRP** | C-reactive protein |
| **CSF** | Cerebrospinal fluid |
| **ECMO** | Extracorporeal membrane oxygenation |
| **HUS** | Hemolytic uremic syndrome |
| **IPD** | Invasive pneumococcal disease |
| **MIC** | Minimal inhibitory concentration |
| **PD-1** | Programmed cell death 1 protein |
| **T-Ag** | Thomsen-Frieden antigen |

### Funding
The authors received no funding for this work.

### Competing Interests
The authors declare there are no competing interests.

### Author Contributions
- Nan-Chang Chiu conceived and designed the experiments, performed the experiments, reviewed drafts of the paper.
- Hsin Chi conceived and designed the experiments, contributed reagents/materials/analysis tools, reviewed drafts of the paper.
- Chun-Chih Peng analyzed the data, reviewed drafts of the paper.

- Hung-Yang Chang, Daniel Tsung-Ning Huang, Lung Chang and Wei-Te Lei contributed reagents/materials/analysis tools, reviewed drafts of the paper.
- Chien-Yu Lin performed the experiments, analyzed the data, wrote the paper, prepared figures and/or tables, reviewed drafts of the paper.

## Human Ethics

The following information was supplied relating to ethical approvals (i.e., approving body and any reference numbers):

This study was approved by the Ethics Committees of MacKay Memorial Hospital, Taiwan (IRB No.: 16MMHIS035e).

## Data Availability

The raw data has been supplied as a Supplementary File.

## Supplemental Information

Supplemental information for this article can be found online at http://dx.doi.org/10.7717/peerj.2941#supplemental-information.

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
