# Peer review of "Retrospective study of prognostic factors in pediatric invasive pneumococcal disease"

_PeerJ, doi:10.7717/peerj.2941_

## Round 0.1 · original submission · Major Revisions

Dear Authors, according to major concerns raised by the reviewers (in particular Reviewers 3 and 2), this manuscript must be reviewed very carefully before it can be further considered for publication in PeerJ

·

Basic reporting

No Comments.

Experimental design

No Comments.

Validity of the findings

No Comments.

Additional comments

The positive significant tests (Leukopenia, Thrombocytopenia, elevated CRP, ECMO use) seem to be less or more general tests that are not specific for a postive pnumococcal infection, so that they could have benn influenced by other associated factors?!

Reviewer 2 ·

Basic reporting

No Comments

Experimental design

The aim of the study could be helpful in identifying severe forms of invasive pneumococcal disease. However, the methods do not describe patient selection criteria to be included or excluded.

Validity of the findings

Although the statistical analysis were performed correctly, the findings are not valid for the following reasons:
1. patients with underlying diseases are included
2. it is not clear when laboratory tests were performed
3. the sample size is small

Additional comments

The patient selection criteria in methods are an essential element for the validation of results and this is a major limitation of the manuscript.

·

Basic reporting

I appreciate the opportunity to review this manuscript. The authors performed a retrospective analysis of pediatric patients with invasive pneumococcal disease. They found a mortality of 10% and identified a few factors potentially influencing survival.

The manuscript is well written and easy to read.

Experimental design

While I agree with the authors regarding the clinical relevance of pneumococcal disease, I have a few issues with the manuscript:

Patient selection:
The authors selected pediatric patients with certain diagnoses and analyzed cases with positive culture for pneumococci. How many cases were analyzed? 50 patients with invasive pneumococcal disease were identified. This number needs to be put into proportion regarding the total number of patients.

Patients were included based on clinical diagnoses and subsequently analyzed for a positive culture result. According to the authors definition, invasive pneumococcal disease is defined by positive culture. Possibly, the authors missed patients with positive culture results but a clinical diagnosis divergent from the ones screened for in the present study.

What was the definition for a pediatric patient? I noticed that the oldest patient was 17 years old. While legally a minor in most countries, pathophysiology is probably closer to an adult in this patient.

Apparently, patients were included when a culture result from blood, CSF or pleural fluid culture was positive for pneumococci (P9L83). A pleurocentesis is necessary to obtain pleural fluid. This was probably not performed in all patients with suspected pneumonia. How was pneumococcal pneumonia diagnosed?

Urine pneumococcal antigen was also included. Did the authors include patients with positive urine antigen but negative cultures? The BinaxNOW test used by the authors has a moderate sensitivity with good specificity.

Validity of the findings

Statistical analysis:
The authors analyzed a multitude of variables. Multivariate testing was performed for variable with a p < 0.05 in univariate analysis. None of the variables analyzed yielded a significant result in multivariate analysis. Thus, most of the conclusions need to be rephrased and expressed more carefully (the authors claim that leukopenia, thrombocytopenia, and high CRP levels were associated with mortality, however this apparently was not true in multivariate analysis).

Additional comments

Summarizing, the authors examined a patient cohort of unknown size for invasive pneumococcal disease. While the topic is interesting, the manuscript suffers from a possible bias in patient selection and weaknesses in statistical analysis. The conclusions are for the most part not justified since the multi-variate analysis showed no significant risk factors for mortality.

·

Basic reporting

No Comments

Experimental design

No Comments

Validity of the findings

No Comments

Additional comments

The article is aimed to survey the clinical manifestations and laboratory results of invasive pneumococcal disease and to identify the prognostic factors of mortality. The title is “Retrospective study of prognostic factors in pediatric invasive pneumococcal disease”.
1. This study is a retrospective study.
2. Several factors influence the outcomes of the study. Some limitations might be occurred.
3. The study is conducted in a teaching hospital. The results of the study might be different in the other hospitals.
4. Finally, please recommend the readers “How to apply this knowledge for routine clinical practice?”.

---

## Round 0.2 · Minor Revisions

Please discuss the necessity of considering other patient-specific factors, when conducting a study of prognostic factors in infectious diseases; for example genetic factors, e.g. polymorphisms within the PD-1, CD14 and TLR4 genes are associated with disease progression and outcome of patients with sepsis. Please seek out suitable studies in this field which you could cite.

Reviewer 2 ·

Basic reporting

No comment

Experimental design

No comment

Validity of the findings

No comment

Additional comments

The revision has improved the article. However the results are not statistically significant in multivariate analysis and this reduces the validity of the found indicators.

·

Basic reporting

No Comments

Experimental design

No Comments

Validity of the findings

No Comments

---

## Round 0.3 · accepted · Accept

After adequately discussing potential con-founder and factors that may impact IPD progression, this ms is now ready to be published in PeerJ.